# Cross-Sectional Survey of Musculoskeletal Disorders in Workers Practicing Traditional Methods of Underground Coal Mining

**DOI:** 10.3390/ijerph17072566

**Published:** 2020-04-09

**Authors:** Madiha Ijaz, Sajid Rashid Ahmad, Muhammad M. Akram, Steven M. Thygerson, Falaq Ali Nadeem, Waheed Ullah Khan

**Affiliations:** 1College of Earth and Environmental Sciences, New campus, University of the Punjab, Lahore 54590, Pakistan; sajidpu@yahoo.com (S.R.A.); drakrampu@gmail.com (M.M.A.); Waheedkhan008@gmail.com (W.U.K.); 2Department of Public Health, College of Life Sciences, Brigham Young University, Provo, UT 84604, USA; steven.thygerson@byu.edu; 3College of Statistical and Actuarial Sciences, University of the Punjab, Lahore 54590, Pakistan; nadeemfalaqali26@gmail.com

**Keywords:** Coal mine, upper limb, lower limb, RULA, discomfort, task-based prevalence of pain

## Abstract

*Background:* In subcontinental underground mines, coal mining is carried out manually and requires many laborers to practice traditional means of coal excavation. Each task of this occupation disturbs workers’ musculoskeletal order. In order to propose and practice possible ergonomic interventions, it is necessary to know what tasks (drilling and blasting, coal cutting, dumping, transporting, timbering and supporting, loading and unloading) cause disorder in either upper limbs, lower limbs, or both. *Methods:* To this end, R-programming, version R 3.1.2 and SPSS, software 20, were used to calculate data obtained by studying 260 workers (working at different tasks of coal mining) from 20 mines of four districts of Punjab, Pakistan. In addition, a Standard Nordic Musculoskeletal Questionnaire (SNMQ) and Rapid Upper Limb Assessment (RULA) sheet were used to collect data and to analyze postures respectively. *Results:* In multi regression models, significance of the five tasks for upper and lower limb disorder is 0.00, which means that task based prevalence of upper and lower limb disorders are common in underground coal mines. The results of the multiple bar chart showed that 96 coal cutters got upper limb disorders and 82 got lower limb disorders. The task of timbering and supporting was shown to be dangerous for the lower limbs and relatively less dangerous for the upper limbs, with 25 workers reporting pain in their lower limbs, and 19 workers reporting pain in their upper limbs. Documented on the RULA sheet, all tasks got the maximum possible score (7), meaning that each of these tasks pose a threat to the posture of 100% of workers. The majority of participants (182) fell in the age group of 26 to 35 years. Of those workers, 131 reported pain in the lower limbs and slight discomfort (128) in the upper limbs. The significance value of age was 0.00 for upper limb disorder and was 0.012 for lower limb disorder. Frequency graphs show age in direct proportion to severity of pain while in inverse proportion with number of repetitions performed per min. *Conclusions:* All findings infer that each task of underground coal mining inflicts different levels of disorder in a workers’ musculoskeletal structure of the upper and lower limbs. It highlighted the need for urgent intervention in postural aspects of each task.

## 1. Background

Occupational health and safety problems prevail in every industry, especially in those industries that rely heavily on manual means of handling and performance [1]. Work related musculoskeletal disorder (MSD) are one such ergonomic disorder [2,3]. Technical development and advancement has created many means of ergonomic intervention to reduce occupational hazards of this kind [4]. Despite these technical improvements, a large number of workers are still falling prey to work related MSDs. [5]. This rising number adds economic burden to the industry [6] despite serious attention from respective legislative groups [7].

A disruption in the musculoskeletal order of a person is classified as a musculoskeletal disorder and abbreviated as MSD. MSDs account for any inflammation, tendon, cramp, or poorly functioning muscle, nerve, bone, or joint [8]. When a worker develops MSDs due to job tasks performed at work, it is referred to as a work related MSD and abbreviated as WRMSD [9]. Different types of jobs pose different types of WRMSDs; however, upper limb disorders are the most common in many industries [10]. Upper limb, lower limb, and spinal cord, with addition of the neck, are broad divisions of WMSDs [11]. However, there are many industries with job tasks, that when performed, target multiple musculoskeletal regions of the body [12]. These once short-term discomforts, when repeated over time, could lead to permanent disability [13].

The mining industry has been recognized as one of the most hazardous industries for workers’ health. This factor demands the taking of special measures to reduce worker risk [14]. Underground mines in comparison with surface mines, are especially dangerous as the severity of a hazard is much higher the majority of the time. Be it a fall, collapse, gas emissions, or blast, the health and safety of each worker is highly at risk in this sector [15]. 

In Pakistan, coal mining is very important and is a large economic contributor. It was estimated, that Pakistan’s coal reserves totaled to about 187 billion, Thar area of Sindh Province taking 175 billion tons from this total [16]. In the salt ranges of Punjab Province, reservoirs of coal are considerable and are actively being mined [17]. Although, increased mining of the overlaying coal layers has led to a preference of underground mining [18].

The type of underground coal mining as well as the tasks employed vary widely from mine to mine, i.e., common types of underground coal mining include: room and pillar mining, vein mining, shrinkage stoping, sublevel open stoping, vertical cater retreat, and cut and fill stoping [19]. The common tasks involved are drilling and blasting [20], coal cutting, dumping to the pass-by, transporting to the outer surface, and timbering and supporting [21,22]. Loading and unloading is the task common to all mines.

Drilling and blasting, the first of the tasks, is done to quickly and effectively split and break apart rock to allow for easy access to coal seams [23]. This task requires a constant repetition of heavy manual lifting [24]. Coal is extracted from coal seams, both manually and mechanically, and is one of the most important tasks of underground coal mining [25,26]. Timbering and supporting is the step after coal cutting, in which workers use timbers and other such material to prevent the falling of rock strata-overlying coal seams [27]. Loading and unloading of the mined coal is done either to transport the coal out of the tunnel or to send it to the market for sale. 

Upper Limb WRMSDs are one of the most common types of MSDs. In Europe alone they account for 45% of all WRMSDs [28]. Drilling and blasting, coal cutting, dumping, transporting, timbering and supporting, and loading and unloading are the tasks of underground coal mines, which lead to Upper Limb WMSDs, shortly abbreviated as UL-WMSDs. The bending and twisting of the upper limbs in awkward postures, excessive repetitions, weight of the cutter, and certain ways of hammering in narrow spaces, jerks and adds pressure to the musculoskeletal order of workers’ bodies [29]. These activities cause immediate harmful effects to the upper limbs.

In the wake of this existing situation, it is found that no published study particularly addresses the role of each task (drilling and blasting, coal cutting, dumping, transporting, timbering and supporting, and loading and unloading) of underground coal mining in causing pain in the upper and lower limbs. This study calculates the impact of each task on pain in upper and lower limbs of workers. Once identified, the most hazardous tasks can be intervened at possible scale. Moreover, some other occupational and physical factors of workers are analyzed for the pain. 

### Research Questions

(1)What role do various occupational factors (tasks, routine, experience, etc.) in combination with a person’s personal characteristics (age, Body Mass Index (BMI), previous injury, etc.) play in the perception of pain, at different levels of job task frequency and severity, in the upper and lower limbs?(2)Which of the two limbs disorder is most prevalent?

## 2. Methods

### 2.1. Study Area 

Four districts: Chakwal, Jehlum, Khushab, and Mianwali, were selected for the study. Five mines were selected from one district totaling 20 mines from all four districts of the study area. These twenty mines were underground coal mines. Each mine was visited three times to get the questionnaires filled and to observe the postures to score the Rapid Upper Limb Assessment (RULA) sheet to score each task of underground coal mining.

### 2.2. Design of Criteria and Participant Selection and Variables 

#### 2.2.1. Selection Criteria Based on Walkthrough Survey 

During the walk-through survey, we obtained the number of workers that would be involved in each task of the process. We calculated that almost every mining site had the same percentages of workers at each task, with 45% to 46 % of workers at coal cutting, 15% to 16 % at dumping, timbering, and supporting, and 7% to 8% of workers at transportation, drilling, and blasting and loading.

#### 2.2.2. Proportional Allocation to Select Population Sample and the Main Characteristics of Each Sample 

Table 1 gives the detailed selection of study participants from each mine and from each task. From each mine, we gathered a group of 13 workers; this group was in proportion to the percentage of workers that were involved in each task of the mining process, and all mines of the study area followed the same six work tasks with allocation of same percentage of the workforce to each work task separately. One group of 13 workers was obtained from each of the 20 mines, totaling 260 workers. Every group of 13 workers comprised of one from drilling and blasting, six from coal cutting, two from dumping, one from transporting, two from timbering and supporting, and one from loading and unloading. These driller and blasters, coal cutters, dumpers, transporters, timbering men and loaders and unloaders were fixed to their tasks. There was no rotation or shuffling of tasks throughout their respective work experiences.

The daily manual handling of work and material by the sampled population varied according to the task. The typical coal cutter (called gaintee, in local terms), weighing 3.5 kg, was used for coal cutting. A mine tub (called chakra, in local language) with capacity to carry 1 ton of coal, hand trolley (of standard size), and manual wheeled-haulage to be dragged on pre-lain track with capacity of 2000 kg of coal, or donkeys, are used to transfer the extracted coal from the seam to pass-by or to the surface. A shovel (called bailcha, in local terminology) weighed 3.5 kg, and was used for loading coal to the vehicle, donkey, mine tub, or hand trolley. A pneumatic drill machine (weighing 25 kg) was used for drilling and it was operated by one worker at a time. A local made hand saw (weighed 0.3 kg) and peeler (called taisa, in local terms) weighed 1 kg, and was used by workers for timbering and supporting. 

The number of repetitions performed by every worker for each task is calculated based on the use of their respective tools or movement of relevant body parts to perform the task in one min. For example, the repetitions of a coal-cutting worker would count the number of times the worker hammered the coal seam with the cutting tool in one min. In the questionnaire of that worker, it would be the simple count/min, but in the RULA sheet, the task of coal cutting would be scored by taking the weight of the cutter into account. Thus, the score of the left arm, shoulder, and wrist would be lower than the right because all workers are right-handed. Likewise, the repetitions of the workers of dumping and transporting tasks would be calculated by counting the number of times they bend and raise their upper trunk (with the weight of already mined coal piled onto the jute sheet by the coal cutting workers who spread this sheet after every 10 to 15 kg cut of coal), to fill the haulage, hand trolleys, or sacks hanging on the backs of donkeys. The task count for drilling and blasting is of the number of times a driller jerks from the pressure of a pneumatic drill machine in one min. To mark on the RULA sheet, this task would be scored based on the angle of shoulders, upper trunk, and legs while taking the weight of the machine into account. Workers who load and unload coal use shovels. The repetitions would count of number of times they bend their upper trunk, fill the shovel with coal, and throw it to or from the surface required to be loaded or unloaded, respectively, per minute. Similarly, the count of number of repetitions of timbering and supporting workers would be the number of times they run the hand saw and peeler (to prepare timber for support in the mined wings) in one min or the number of times they hammer these timbers to fill in the coal seam, which was cut by the coal cutters. On the RULA sheet, this task would be scored based on the angles of upper and lower body parts, the number of repetitions/min, and weight of the hand saw, peeler, and hammer. 

#### 2.2.3. Variables 

Dependent variables included the discomfort in upper and lower limbs, work performance, and frequency of pain, whereas the tasks (drilling and blasting, coal cutting, dumping, transporting, timbering and supporting, and loading and unloading) were the main independent variables. Other predictors of the research included age, working hours, working months/year, and the number of repetitions per min (of exertion of muscle carrying weight). 

### 2.3. Modified Nordic Musculoskeletal Questionnaire (NMQ)

We used NMQ modified, according to the objectives of the study. Classification of right and left parts of limbs were also added. Section one of the questionnaire addressed information, such as name and address of company, operational hours, and total number of employees hired for each task of the mining process. Questions that addressed their age, height, weight, tasks, hours worked per day, and work experience were asked in Section 2 of the questionnaire. The number of repetitions per min of job tasks, injuries obtained outside of work, level of fatigue at the end of the day, etc., were asked in section three. In the fourth section, workers were asked if they currently felt pain in their musculoskeletal system. Their responses were limited to Yes/No. Only those workers who responded “Yes” were included in the study population and were asked to respond to further questions. These further questions asked about pain in the neck, shoulders, arms, elbows, hands/wrists, hips/thighs, knees, and ankles/feet. Question involving the frequency of pain in those areas was also asked in this section. The options for frequency of pain included: within the last couple of days, last seven days, last six months, and last 12 months. The severity of pain was also asked with options of: a bit, rather, severe, and very severe. 

A questionnaire was filled out by the researchers in a structured interview [30]. The reliability and validity of Nordic questionnaires is reported by Palmer to be α = 0.63 − 0.90 and the specific index to be 0.73 to 0.94 [31]. To highlight the particular steps involved in work related musculoskeletal disorders, the Nordic questionnaire placed special focus on musculoskeletal order [32]. 

### 2.4. Rapid Upper Limb Assessment (RULA) for Postural Analysis 

In addition to NMQ, RULA sheets were used to check the level of risk involved at each task of underground coal mining. RULA is a survey method used for ergonomic investigations of occupational musculoskeletal disorders [33]. The RULA sheets are divided into three tables that are guides for limb assessement. Table A in the sheet is for arm and wrist analysis. In step 1 to 4, based on the given positions on the sheet, a researcher will mark upper arm, lower arm, wrist, and wrist twist in the respective boxes. In step 5 to 8, based on the number of repetitions performed, and muscle load, a researcher will add up qualifying scores and calculate final values to enter in the related column of Table C. Table B involves the positions of the neck, trunk, and legs during work. The researcher matches the posture of the worker to the posture of the neck given in the sheet, neck, is scored in step 9. Trunk posture is scored on the basis of deviation from hips to shoulders and is scored in step 10. In step 11, a supported leg scores 1 and a non-supported leg scores 2. In step 12, values from the boxes are marked in table B. Based on the values obtained from step 13 and 14, which involved repetition and muscle load, the researcher in step 15, calculates the final score to be marked in the respective column of table C. After this the intersect value of both step 8 and step 15 is located in table C and becomes the final Score of RULA sheet. This final score is interpreted according to the level of risk involved and modification required. This tool requires no special equipment or tools, only a sheet, pencil, and an observer to assess the postures of the neck, trunk, and upper limbs ,along with muscle function and the external loads experienced by the body [34]. 

### 2.5. Addressing Potential Worker Bias

To avoid bias from workers’ self-reported data, we enlarged the sample size and had researchers rate each activity on the Rapid Upper Limb Assessment sheet. 

### 2.6. Statistical Analysis 

For data analysis, we used R-programming version R 3.1.2. Multiple Regression Models are drawn based on analysis of the data using R-programming. SPSS, 20 software (SPSS Inc., Chicago, IL, USA), is used to draw 1 Multiple Bar Chart and 2 Frequency Plots to show some relation between certain personal and occupational factors. 

## 3. Results 

Reporting of discomfort in both upper and lower limbs by workers from each task was not the same. The pain was different for different working steps of underground mining of coal. From physical traits, age has a great role in this pain. Occupational factors including working months/year, experience, working hours/day, and number of repetitions were also included in this analysis. To find an association between pain in each of the limbs, multiple regression models are drawn separately and significance of several of occupational and personal factors to cause this pain are checked in these models. Moreover, to find the most prevailing limb disorder, an MSD comparison between upper and lower limbs is drawn. Rapid Upper Limb Assessment (RULA) sheet is also used to find the postural risk involved in each work step of underground coal mining.

### 3.1. Means of Physical and Occupational Traits of Workers 

Mean values of different parameters of workers are listed in Table 2. On average, the workers were young and started working at an early age, which enabled them to obtain years of experience at a young age. Average repetition of task performance (counted during task performance by each participant) was 26 times per min with the standard deviation value 9.49, which is high because the number of repetitions performed by 260 workers was different from each other; however, it was somehow equal in workers performing similar tasks. High standard deviations were also seen for BMI (6.515) and number of working h/day (5.23). 

The mean value of BMI is 27 because most of the workers were of shorter stature and carried more weight, which put them into the overweight category. The mean value of calculated age is 27.05 years, while mean of working experience is eight years. They live on mines and opt out for long work shifts to earn extra money. Workers used to work for about 13 hours a day (with only a lunch break of half an hour) and about 8 months a year (which does not depend on seasons, but on the requirements at home), on average. They take off from work, only for couple of months, to visit their distant hometowns. Average severity of pain, which workers reported, falls under category 2.86, which is categorized as severe pain in the questionnaire. 

### 3.2. Reported Pain in Upper and Lower Limbs from Different Personal and Occupational Factors 

The first research question is answered in this section. Results obtained from the questionnaire showed that reporting of pain in both limbs differed according to the different level of task, work performance, age, etc. Six parameters, related to the pain in the upper and lower limbs, are calculated in Table 3. In addition, this table shows the grouping of workers according to age, performance of repetitions, work experience, and working h/day (in one-shift, i.e., no part time or double shift). The repetitions were counted and varied from one worker of a task to another worker of the same task. It is difficult to give repetitions of task performance of 260 workers separately. However, the grouping shows that 93 workers performed the same number (20 times/min) of repetitions and another 90 workers performed the same number (30 times/min) of repetitions. Only four workers went beyond the 50 times of repetitions/min.

Each parameter is subdivided into different groups, according to the data obtained from the questionnaire. Strength of workers is divided into different groups based on difference of age, experience, daily working h, and number of repetitions being performed per min. Underground cutting of coal is divided into six tasks. Pain reported from each task by each respondent is calculated in separate columns. To represent the total number of workers in each subcategory, a separate column is given in the table. 

Out of 120 coal cutting workers, 91 reported pain in the right upper limb (which includes hand, wrist, arm, shoulder) with 73 in the left upper limb, and 82 reported pain in both right and left upper limbs. Workers of this task also gave the highest number of reporting, 63, for pain in both upper and lower limbs at a time. Neck pain was reported in 20 out of 40 workers from the timbering and supporting task. This task reported the highest pain levels for this part of the body. The next highest was 19 workers reporting neck pain from the coal transporting task. There were 181 workers, the maximum number of workers, in age group ii (26 to 35 years of age). Out of these 181, 131 reported pain in each, right and left, side of the lower limbs (which includes the hip, leg, including knee, and foot), 128 reported pain in the right side of the upper limb, and 127 reported pain in neck. 

Table 3 also shows a range of total months for which workers work in a year. The minimum number of working months is four while the maximum number is 12 months. These working months are without any holiday. In a bid to earn the maximum money to send to family, workers do not take any rest days. A total of 170 workers work from 7 months to 9 months, compared to 88 workers from 9 months to 12 months. Out of these 88 workers, 44 workers reported pain in the upper limb and 57 in the lower limb. The number of repetitions per min being performed by workers also have important results. The lowest count of repetitions per min is 10, while the maximum count is 50. The highest number of workers, 93, perform 20 times the repetitions per min; out of these, 55 reported pain in the neck, 63 reported pain in both right and left sides of the lower limb, while 62 reported pain in the right side of the upper limb. 

#### 3.2.1. Frequency and Severity of Pain

Table 4 summarizes the results of pain, vis a vis its occurrence/frequency and severity of pain. The maximum number of workers reported that they had felt pain in the last week, with 55 workers having ‘rather severe’ and 54 workers having ‘severe’ pain. 

#### 3.2.2. Level of Significance of Occupational and Personal Factors to Inflict Disorder in Upper Limbs of Coal Miners

Every task involved in the process of coal mining requires exertion of different body parts. Such variety of tasks targets different parts of the body to trigger pain and discomfort. Model A (in Table 5) proves the significance of task, age, working months/year, experience, and number of repetitions for discomfort in the upper limb of workers. 

Model A: Multiple regression model for work related upper limb disorder. 

Results find a strong level of WRULMSD triggered by different characteristics of the worker and the work involved in underground mining of coal. Significance value of age and tasks is 0.000, which is <0.005 for pain in the upper limbs of underground coal mining workers. Working months, number of repetitions, and experience of workers also contribute toward inflicting pain in the upper limbs of workers. 

#### 3.2.3. Level of Significance of Occupational and Personal Factors Inflicting Lower Limb Disorders 

The association between lower limb disorder and age, working months in a year, and tasks performed by workers in the process of underground coal cutting is also calculated. Model B (Table 6), shows the level of significance of different parameters to cause lower limb disorder. 

Model B: Multiple regression model for work related lower limb disorder. 

The tasks and severity of pain have significant values of <0.005 as compared to the other parameters shown in the model. The *p*-value of work experience, number of repetitions/min, and age are 0.004, 0.009, and 0.017, respectively.

### 3.3. Comparison of Prevalence of Disorder between the Upper Limb and Lower Limb 

To find which of the two limb disorders prevails, a comparative bar chart is drawn (Figure 1). The prevalence of upper limb disorder is most common as compared to that of lower limb disorder. The total number of workers involved in each task is given in Table 1 of the previous section, i.e., methodology. This bar chart also shows the overlapping of reporting of disorder in both limbs at a time. 

The maximum reporting of the upper limb, lower limb, and both limbs is from coal cutting, with 91 workers reporting pain in upper limbs, 82 in lower limbs, and 62 reported pain in both limbs. The second highest reporting for the upper limb is from dumping, with 30 reports, while the second highest reporting for the lower limb is from timbering and supporting, with reporting from 29 workers. 

### 3.4. Other Important Miscellaneous Findings 

#### Role of Age in Job Performance and Severity of Pain at Underground Coal Mines 

To study the role of age in work performances and severity of pain, three graphs (Figure 1, Figure 2 and Figure 3) are drawn. The 260 study participants were divided into three age groups; 16–25, 26–35, 36–45. The number of repetitions decrease with increase in age of workers. Severity of pain increases with increase in age. From Figure 2, the respondents above age 35 report high levels of pain in relevant parts of the body, asked in the NMQ. From Figure 3, workers in lower age groups perform a high number of repetitions, and vice versa. 

### 3.5. Evaluation of Each Task Using RULA 

Postural analysis of all workers occupied at their respective task was carried out using Rapid Upper Limb Assessment (RULA) sheet and results (see Table 7). 

Posture in section A of the RULA sheet consists of arm and wrist analysis, and is called Posture A in many studies. Having scored the posture of each worker according to the steps described in methodology, posture A (right) received a score of 6 to 9 from table A, depending upon the workers’ position at work and the task they perform. Repetitions were involved >4, so score 1 was added in step 6 (i.e., Muscle Use Score). Underground coal mining of the study area is based on manual techniques; workers carry heavy loads of their tools or trolleys (coal cutter, a mine tub, shovel, hand trolley, pneumatic drill machine, hand saw, peeler, etc.), so a score 2 is added in step 7. The values obtained in step 8, from calculations of step 5, 6, and 7, are the final score (with a range of 8 to 12) of arm and wrist analysis, and is marked in table C to get the grand RULA score. Since the last value in the related column of table C is 8+, so all values are marked at 8+ for each worker. Similarly, final values calculated for Posture A (left) are in range of 6 to 10. For posture A (right), the maximum score is 12 for 86% of workers of coal cutting and 77% of workers from transportation, and 50% from loading of coal to vehicle. The highest value for posture A (left) is 10 for 100% of workers from coal cutting and 48% of workers from transportation.

Section B of the RULA sheet is for neck, trunk, and legs analysis. The right and left of posture B is not possible because of the neck and trunk and, in the sheet, legs are to be analyzed collectively. Score for this posture, from table B, is in range of 5 to 8 for all workers in different tasks. Muscle use score, step 13, for all workers was 1, and force/load, step 14, was 0, 1, or 2. The final values for posture B were in the range of 7 to 10, whereas the maximum value for this posture in table C is 7+, so all of these obtained values were marked at 7+ in the table. The highest score for this posture is 10 for 72% of workers from timbering and supporting and 59% from drilling and blasting. The grand RULA score for all workers (100%) of every task is 7, which is the maximum score.

## 4. Discussion

The first research question (i.e., which of the occupational and personal factors cause pain in upper and lower limbs?) is answered in Table 2 and Multiple Regression Models 1 and 2. From the results, it is found that among occupational factors of pain in the upper and lower limbs, tasks are at the top of the list. Coal cutting hires the most workers and is the cause of pain, largely in the upper limb and relatively less large in lower limbs. Likewise, the other six tasks involved in coal mining are harmful for the limbs of workers. It is important to note from the table that reporting overlapped—and many workers reported pain in both upper and lower limbs at a time. This necessitated further analysis of the results and resulted in multiple regression models to check the significance. To find the link between occupational factors and pain in limbs of mining workforce, Multiple Regression Models are drawn. Model 1 and Model 2 show the strong influence of occupational factors, including work experience, working months/year, working h/day, and number of repetitions/ min, over pain in both limbs. The reason of this strong influence lies in occupational requirements of mining. Underground mining of coal is recognized as one of the most hazardous occupations throughout the world [35]. It provides workers a very narrow space to work [36]. They have to cut coal while sitting, laying on their backs or stomachs, kneeling, or bending at awkward postures. Drilling and blasting, coal cutting, and timbering and supporting require hammering of tools. Constant hammering results in a repetitive strain on the body [37]. Moreover, they have to hammer their heavy cutter for long working h. When the risk factors of force and repetition are observed in a task, the risk of injury greatly increases compared to a task that just has one of the risk factors without the other. In coal mining, tasks requiring high levels of force combined with repetitive cycles explains the high number of MSDs seen in workers. The impact of daily working hours has a great role in causing permanent disability in workers [38]. Installing and maintaining timbering also poses health hazards, as workers frequently work in low coal seams and are in constant awkward postures [39]. Such situations create upper limb disorders.

For lower limbs, reporting of pain in Table 2 differed. Bending down, lifting objects, or dragging material are common activities causing discomfort in lower limbs [40]. The mine surface is uneven and inclined at great angles, which makes lower limbs of transporters vulnerable. Risk of knee disorder increases when workers walk with loads, and with their trunks inclined forward [41]. The loading of coal is carried out in standing positions. It involves moving the back up and down. The movement of the back is coordinated with hip and spine joint movement [42]. Coal cutting in very narrow seams forces workers to fold their legs, sit on their hips or feet, and bend their knees awkwardly [43]. Sometimes workers have to fold their legs to be able to work in a very narrow space. Moreover, at times they have to put all of their body weight on their legs and knees, as shown in plate 4. Moreover, any coordinated movement in the upper or lower back initiates pain in the lower limb [44]. Such circumstances become immediate causes of minor or major disorder in lower limbs.

The second research question (i.e., which of the two limbs- upper and lower- disorder is most prevalent?) is answered in Figure 1. Here, we see that the upper limb disorder is most prevalent. The reason is that activities involved in underground coal mining are multi-directional and most of these activities focus on the upper limbs. Based on RULA postural analysis, the lower limbs are much less affected compared to the upper limbs.

The affected limb is based on the excessive use [45] and in the case of underground coal mining, results tell that excessive use is of upper limb parts. From the values, it is clear that drilling and blasting for coal has a lower level of impact on lower limbs as fewer workers are reporting pain because of this activity. Whereas the transporting of cut coal has a stronger influence over lower limb disorder with the maximum value of reporting against the tasks. Among the workforce, use of upper body parts is most common. The repetitions per min also add pressure of upper limbs. Shoulders and hands are used at long stretches without any break. Therefore, insufficient break time increases the burden on upper body parts.

Lower limb exertion is relatively less. When shoulders, forearms, and elbows work, then hips, knees, and feet are usually resting. The use of lower limbs in underground mining of coal is not as continuous as of upper limbs. Studies show that in coal mining, it is common for the lower limbs to be at rest when the upper limbs are working. [46]. However, it does not mean that lower limbs are free of any kind of disorder in underground mines. Indeed, knee pain is as common as shoulder pain; feet are as painful as hands. Both problems exist, but when a comparison is drawn between the two, upper limb disorder stands on top.

In the model applied, three parameters are found to have considerable role in causing discomfort in the upper limbs of workers. According to the standards, the *p*-value should be less than 0.005 [47]. In the table, given above, the *p*-value of all three parameters was <0.005, which means they are strongly significant to become the cause of upper limb disorder in the workforce of underground mining of coal.

Age is a risk factor for body pain and number of repetitions performed per min. Repetitions of a task per min is influenced by age, as the aged workers lack sufficient energy to repeat the maximum numbers [48]. Young workers perform more energetically, with the maximum number of repetition to produce the maximum output [49]. Young workers are more oriented toward earning high wages as compared to the aged workers, so they perform the maximum job tasks in a day [50]. Similarly, aged workers feel more pain in the limbs at the end of the shift. [51]. It implies that with an increase in age, pain in the limbs increases, and vice versa.

Contrarily, other parameters that were applied in the model were of less significance to become a reason of discomfort in the upper limbs of workers. Body Mass Index (BMI), work experience, and working hours in a day were less significant to have an impact on upper limbs. For this reason, such parameters are not included in any statistical model.

The RULA scoring is based on number of repetitions, carrying force, and angle of movement from the reference point. Posture, number of movements, working at a stretch, and the force involved are main elements to be considered when using the RULA assessment sheet [34]. In this method, a sheet is used, on which angular movements of parts of the upper Limb, shoulder, elbows, wrist, and hands are displayed. A scoring table is given with special instructions to add to the score if the number of repetitions and weight increase [52].

As compared to other activities, transporting mined coal has some effect on the upper limbs. It is most related to the collection of excavated coal and delivering it to the dumping sites. Exertion of upper body parts is done in transporting, but this task is less repetitious. Research shows that allowing even short breaks reduces pressure on the musculoskeletal system, and is always a recommended practice [53].

Drilling and blasting also prompts discomfort in the upper limbs. It involves high repetition tasks and requires forceful exertion of upper limbs. Coal cutting is the most hazardous task as it involves carrying heavy tools and sitting or lying in awkward postures. It involves jarring and jolting when striking the cutter or other tools to the coal seam [54]. RULA helps in posture assessment with special focus on number of movements per min [55]. At some mining sites, coal cutting requires workers to lay on their bellies for more than 8 hours a day to cut the targeted quantity of coal. In such circumstances, the activity becomes more dangerous.

### Limitations of the Study

➢RULA’s score is not sufficient for the postural analysis of mining workers, as we can see from the results that values go up to 12, while the maximum scale on the sheet was 8+ for posture A and 7+ for posture B.➢Several workers were reluctant to share their miseries or they were unable to explain the place with frequency of pain.

## 5. Conclusions

The study demonstrates that the occupation of underground coal mining poses a great threat to the musculoskeletal order of its workforce. This study focused on disorders in upper and lower limbs in the Pakistani coal mining workforce. Every work task (drilling and blasting, coal cutting, dumping, transporting, timbering and supporting, loading and unloading) of coal mining leads to musculoskeletal disorders. The coal cutting task emerged as the most hazardous for the upper limbs of workers. However, lower limb disorders were highest in workers of transportation tasks, and minimum in workers of drilling and blasting tasks. Out of these two studied disorders, results highlighted upper limb disorder as the most prevalent. The tasks that were found hazardous for the limbs of workers during regression analysis also received a high score on the RULA sheet. Among the miscellaneous findings, it was found that aged workers reported high severity of pain and were also slow in work performance. All of these prevailing circumstances highlight the urgency of ergonomic interventions (which can be in the form of certain exercises, modification in working postures and tool handling, proper rest time within the shift duration and job rotation), mechanization of coal mining, and proper legislation at the government level. The present study could not practice (on any controlled group) any of the possible ergonomic interventions. Future studies could focus on interventions that can be experimented and suggested for workers of underground coal mines of countries where the occupation is being practiced manually.

## Figures and Tables

**Figure 1 ijerph-17-02566-f001:**
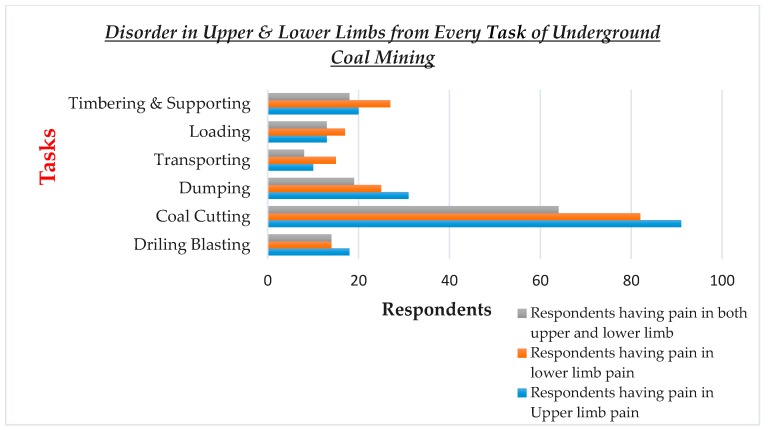
Comparison of pain reported in upper, lower limbs, and then both limbs.

**Figure 2 ijerph-17-02566-f002:**
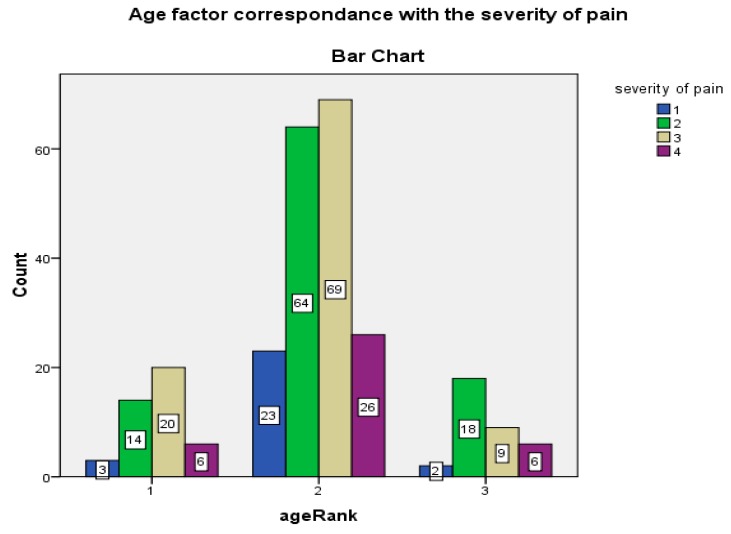
Relation between workers’ age and ability to perform number of repetitions/min.

**Figure 3 ijerph-17-02566-f003:**
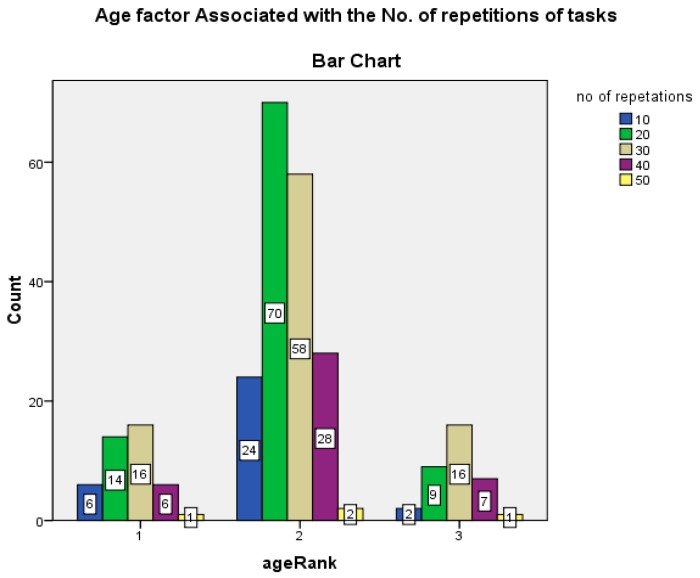
Age of workers inflicting different levels of pain.

**Table 1 ijerph-17-02566-t001:** Participants selected from each mine and each task.

	Mines	M1	M2	M3	M4	M5	M6	M7	M8	M9	M10	M11	M12	M13	M14	M15	M16	M17	M18	M19	M20	Total No. of Workers Involved in Each Task
Tasks																						
**Drilling and Blasting**		1	1	1	1	1	1	1	1	1	1	1	1	1	1	1	1	1	1	1	1	20
**Coal Cutting**		6	6	6	6	6	6	6	6	6	6	6	6	6	6	6	6	6	6	6	6	120
**Dumping**		2	2	2	2	2	2	2	2	2	2	2	2	2	2	2	2	2	2	2	2	40
**Transporting**		1	1	1	1	1	1	1	1	1	1	1	1	1	1	1	1	1	1	1	1	20
**Timbering and Support**		2	2	2	2	2	2	2	2	2	2	2	2	2	2	2	2	2	2	2	2	40
**Loading and unloading**		1	1	1	1	1	1	1	1	1	1	1	1	1	1	1	1	1	1	1	1	20
**Sample from each Mine**		13	13	13	13	13	13	13	13	13	13	13	13	13	13	13	13	13	13	13	13	**260**

**Table 2 ijerph-17-02566-t002:** Mean values of different personal and occupational factors.

Descriptive Statistics	Mean	Standard Deviation
Age	27.05	11.6
BMI	27.43	6.515
No of working h/day	12.63	5.23
No of working months	8.43	2.5
No of working years (experience)	8	4.1
No of repetitions/min	25.85	9.49
Severity of pain	2.86	0.891

**Table 3 ijerph-17-02566-t003:** Reported related musculoskeletal disorder (MSD) pain from personal/occupational factors.

Factors	Total Number of Workers	Number of Workers Reporting Pain in Each Section					
		Pain in Neck	Pain in Upper Limb		Pain in Lower Limb		Pain in Both Upper and Lower Limbs
			Right	Left	Right	Left	
**1. Tasks**							
i. Drilling and blasting	20	12	17	6	16	16	14
ii. Coal cutting	120	71	91	73	82	82	63
iii. Dumping	20	13	9	9	15	15	7
iv. Transporting	40	19	30	30	21	21	17
v. Timbering and supporting	40	20	17	17	29	29	17
vi. Loading	20	16	11	8	15	15	10
**2. Age (years)**							
i. 16–25	43	30	31	14	30	30	21
ii. 26–35	182	127	128	119	131	131	95
iii. 36–45	35	20	19	19	22	22	16
**3. Experience in years**							
i. 4–7	44	24	31	17	27	27	25
ii. 8–11	126	58	57	43	61	61	53
iii. 12–15	87	29	39	39	30	30	27
iv. >15	3	3	2	2	3	3	2
**4. Working months in a year**							
i. 4–6 months	2	0	1	1	1	1	1
ii. 7–9 months	170	140	135	126	125	111	117
iii. 10–12 months	88	22	44	27	57	57	41
**5. Working hours in a day**							
i. 4–6 h	73	31	34	30	29	29	24
ii. 7–8 h	104	62	60	51	57	57	49
iii. 9–10 h	71	34	36	36	32	32	27
iv. 11–12 h	12	4	8	8	5	5	3
**6. Number of repetitions per min**							
i. 10 times	32	26	25	19	22	22	10
ii. 20 times	93	55	62	52	63	63	49
iii. 30 times	90	60	61	49	62	62	42
iv. 40 times	41	26	30	26	27	27	19
v. 50 times	4	3	3	3	1	1	1

**Table 4 ijerph-17-02566-t004:** Workers’ reporting on frequency with severity of pain.

Severity of Pain	Response to Question “When Did you Feel Pain Last Time?”			
	Last Day	Last Week	Last 6 Months	Last 12 Months
A bit pain	8	18	2	1
Rather pain	15	55	34	6
Severe pain	27	54	3	3
Very severe pain	13	17	1	3

**Table 5 ijerph-17-02566-t005:** Parameter Estimates (Coefficients).

	Estimate					95% Confidence Interval foe Beta_(s)_	
	Unstandardized Coefficients	S.E	Standardized Coefficient of β	*t*-Value	Sig.	Lower Bound	Upper Bound
Intercept	0.50835	0.14769	-	3.241	**0.000 ***	0.217	0.799
Age	0.01467	0.0041	0.142	3.422	**0.000 ***	0.007	0.024
Working months in a year	−0.00142	0.00764	0.00316	−0.214	0.041 **	−0.011	0.016
Tasks	−0.074253	0.0122	0.0359	−4.214	**0.000 ***	−0.114	−0.031
No. of Repetitions	0.00354	0.00156	1.107	0.423	0.0096	0.0012	0.101
Work Experience	0.0411	0.01201	0.346	0.311	0.0101	0.0155	0.177

* Statistically significant at 0.05 value; ** Statistically significant at 0.10 value.

**Table 6 ijerph-17-02566-t006:** Parameter Estimates (Coefficients).

	Estimate				95% Confidence Interval foe Beta_(s)_	
	Unstandardized Coefficients	SE	*t*-Value	Sig.	Lower Bound	Upper Bound
Intercept	0.6124	0.15114	3.335	0.000 *	0.3334	0.822
Age	0.01458	0.00424	2.526	0.017 *	0.002	0.141
Working months in a year	0.01541	0.00475	−1.387	0.027 **	0.007	0.094
Tasks	0.01214	0.01122	0.581	0.000 **	−0.024	0.044
No. of Repetitions	0.0166	0.00416	1.12	0.009 **	0.004	0.211
Work Exp.	0.0211	0.01234	0.311	0.004 **	0.0132	0.145
Severity of pain	0.01345	0.00654	1.011	0.000 *	0.0112	0.347

* Statistically significant at 0.05 value; ** Statistically significant at 0.10 value.

**Table 7 ijerph-17-02566-t007:** Analysis derived from using Rapid Upper Limb Assessment (RULA) assessment sheet.

Section-Wise Analysis	Drilling and Blasting		Coal Cutting		Dumping		Coal Transport		Loading		Timbering and Supporting	
	RULA Scores (% Age of Workers)		RULA Score (% Age of Workers)		RULA Score (% Age of Workers)		RULA Score (% Age of Workers)		RULA Score (% Age of Workers)		RULA Score (% Age of Workers)	
**Posture A (right)**	**6**	**7**	**9**	**8**	**6**	7	9	8	8	9	7	6
Muscle	1	1	1	1	1	1	1	1	1	1	1	1
Force/Load	2	2	2	2	2	2	2	2	2	2	1	2
Wrist and Arm (right) score	8 (15%)	10 * (85%)	12 * (86%)	11 * (14%)	9 * (32%)	10 * (68%)	12 * (77%)	11 * (33%)	10 * (50%)	12 * (50%)	9 * (30%)	8 (70%)
Posture A (left)	5	5	8	-	6	-	6	7	6	6	7	6
Muscle	1	1	1	-	2	-	1	1	1	1	1	1
Force/Load	1	0	1	-	1	-	2	2	2	1	1	1
Wrist and Arm score (left)	7 (79%)	6 (21%)	10 * (100%)	-	9 * (100%)	-	9 * (52%)	10 * (48%)	9 * (40%)	8 (60%)	9 * (76%)	8 (24%)
Posture B	8	7	8	-	5	6	5	6	6	5	7	7
Muscle	1	1	1	-	1	1	1	1	1	1	1	1
Force/Load	1	1	0	-	1	1	2	2	2	2	2	1
Neck, trunk and Leg score	10 ** (59%)	9 ** (41%)	9 ** (100%)	-	7(48%)	8 ** (52%)	8 ** (61%)	9 ** (39%)	9 ** (45%)	8 ** (55%)	10 ** (72%)	9 ** (28%)
Final score	7 (100%)	7 (100%)	7 (100%)	7 (100%)	7 (100%)	7 (100%)	7 (100%)	7 (100%)	7 (100%)	7 (100%)	7 (100%)	7 (100%)

Note: (*) means that wrist and arm score would be marked at 8 + while ** means Neck, Trunk, and Leg score would be marked at 7 + n the Table C of the RULA sheet.

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
