# Peer review of "Cross-Sectional Survey of Musculoskeletal Disorders in Workers Practicing Traditional Methods of Underground Coal Mining"

_ijerph, 2020, doi:10.3390/ijerph17072566_

Round 1
Reviewer 1 Report
Over all recommendations:
- This is a very important study that needs to get published.
- There are specific corrections that should be made which are identified in the next section.
- Divide this research into at least two publications. The first publications should focus on the results of the Nordic Musculoskeletal Questionnaire (NMQ). In this first paper there needs to be more attention paid to the description of each task that is analyzed. Keep the six tasks (drilling/ blasting, coal cutting, dumping, transporting, loading and timbering/supporting). Decide what order to address each task and keep it consistent throughout the manuscript. Provide a description of each task so that the reader has an idea of the work being performed especially as regards posture, repetition and force.
- Describe the tools and equipment used during each task. Make it clear which worker performs which task. Do workers perform all tasks? Or do workers perform only one task? If they perform a few tasks which ones? How heavy are the tools? How heavy are the loads lifted, pushed or pulled? Identify the duty cycle for each task and indicate how long it takes to complete on full cycle. The researcher/ergonomist has to evaluate posture, force and repetition. These are objective, quantifiable variables and should not be reliant on researcher or worker perception. Workers can report pain, age, length of time working and other variables recorded in the manuscript.
- There needs to be a justification for grouping work in underground and above-ground mines together. The manuscript states “The type of underground coal mining as well as the ‘work stages’ (change to ‘tasks’) employed vary widely mine to mine.” Please justify how all 20 sites can be lumped together. There may be justification for doing that but there needs to be an explanation. Perhaps the six tasks described in the manuscript are similar among all sites.
- Because there is very little data available on mining in Pakistan it is very important to get this initial information published. There are other publications that might serve as a template for doing this such as https://www.ncbi.nlm.nih.gov/pmc/articles/PMC2823665/ where researchers use a questionnaire to evaluate musculoskeletal pain in nurses in Nigeria.
- Once the initial data is published it would be terrific if the RULA methodology is used to further clarify the upper extremity musculoskeletal effects you are describing. RULA really does not evaluate lower extremities directly, only as they provide stability for upper extremity activity. Low back pain and disability may be a better parameter to evaluate in miners which is absent from the current manuscript. But that could be a third publication.
Specific comments on the manuscript:
- Shouldn’t Steve Thygerson be connected to Bringham Young University and Muhammad Akram be connected to the University of the Punjab?
- The paper identifies “work stages.” These are typically called “tasks” in ergonomics. Using the term “task” instead of “work stages” would facilitate understanding what was done.
- “Work stages” or “tasks” are not consistently described throughout the paper. Table 4 (on page 10 of 16) represents 6 tasks: (1) drilling and blasting (2) coal cutting (3) dumping (4) transporter (5) loading and (6) timbering and supporting. The abstract only lists five tasks. Section 2.2.3 also only lists five tasks. Section 2.2.2 lists six tasks. Each time the tasks are listed they are presented in a different order. This is very confusing for the reader. Decide on the order in which the tasks are listed and use it consistently throughout the paper.
- Sections 2.2.1 and 2.2.2 could be combined in a table which would make the material much easier for the reader to understand.
- Section 2.1 states that 20 mines were studied but only 5 were underground coal mines. Were all 20 mines included in the study? If only 5 mines were included in the study why talk about the other 15 mines? Were the surface mines coal mines? Most importantly were all the mines the same? Were there differences in how each mine was organized and how the work was done?
- Table 1: The number of subjects (n) on which the mean is based should be reported. Variance is the square of the standard deviation but not very useful to understand the values in Table 1. Much better to report the standard deviation so that the reader can understand the data better. The mean BMI of the workers in this study is 27.43, which indicates they are overweight. Please check the numbers and the calculations and explain. The mean age reported in Table 1 is 19.8 +5 years which contradicts the statement on the abstract that the majority of participants (182) fell in the age group of 26 to 35 years. Please reconcile.
- Table 2 indicated that all 20 mines were included in the study with each mine doing the same tasks (underground versus surface). Reporting left versus right side for upper and lower limb is interesting but some justification for this bilateral reporting needs to be offered. In many cases there is no difference between right and left sides. When there is a difference some explanation of why should be included. Given the prevalence of back pain why is that not addressed in the study? Are you using leg pain as a surrogate for back pain via the sciatic nerve? This approach needs some explanation.
- Figure 1 reports number of workers but the number of workers in each group were not the same. A better way to represent this data is by percentage of workers who report pain.
- For figure 2 a more easily understood representation would be to divide the workers into age groups and relate the incidence of reported pain by age groups. For example, age groups could include all those in a 5 or 10 year span such as 15 to 20 year-olds, 20 t0 25 year-olds; OR 15 to 25 year-olds, 25 to 35 year-olds and so forth.
- Figure 3 is a very interesting presentation and conveys the point quite well. The severity of pain appears to correlate very well on an individual basis with age. Grouping workers by age (as in Figure 2) would also convey the same information and perhaps make it more easily comprehended by readers.
- Figure 4 on the RULA results is disappointing for several reasons: (1) muscle repetitions are reported as “1” which indicates repeated more than 4 times per minutes. Although technically true it does not reflect the various tasks very accurately. Some of the reported repetitions are significantly higher (see table 2) and this is not discussed in the paper; (2) force is reported without any field measurements. A force gauge can be used for lifting, pushing and pulling and needs to accompany this type of analysis; (3) not sure what reporting percentages mean. Percentages of what? RULA scores are used to characterize risk during specific tasks. Not across an entire gamut of activity. They are useful to identify how risky a task is and to prioritize intervention. The strength of RULA lies in its identification of risky postures. It is clear from the report that force and repetition also play significant roles in the development of musculoskeletal injury in the mining occupation.
- The number of repetitions appear to be self-reported. Table 2 does not identify the number of repetitions associated with each task which is necessary to evaluate the risk.
- No recognition is given to the interaction of force and repetition in determining musculoskeletal risk.
Author Response
See attached document.

Reviewer 2 Report
This article addresses the important issue of musculoskeletal disorders in workers of underground coal mining to know what work stages (drilling & blasting, coal cutting, transporting, timbering & supporting, loading & unloading) cause disorder in either upper limbs, lower limbs, or both. The study employed the popular techniques (SNMQ and RULA) to collect and analyze data. The reviewer has no doubt that the authors have a good grasp of the subject and such techniques. However, I believe that the manuscript needs some revisions.
Introduction
- The novelty and originality of the study has not been mentioned in the manuscript. I think the originality of the article can be, for example, an understanding of what work stages are the main causes of musculoskeletal disorders and what ergonomic interventions are necessary to address them. I recommend the authors to explain the novelty and originality of the study at the end of the introduction section.
- Page 2, paragraph 2, the last sentences need a reference(s).
Methods
- In methods, please explain if you had visits of the cites (mines) or observations of work stages.
- Page 4, second paragraph, add the years for the two citations (Palmer et al.) and (Dawson, Steele, Hodges, & Stewart).
Results
- Table 1: Please add the SD column, Mean and SD are most appropriate.
- Table 2: is it possible to specify working hours in a day includes night, day or both. And working months in which season?
- Please include Standardized Beta in Table A2.
- Please include the legend for upper limb pain in Figure 1, 2, 3
- Figure 1 is redundant because all the info exists in Table 2. I suggest this figure must be changed based on the percentage of respondents for each step, the current figure does not show what you are intending to convey.
- Figure 2 & 3 are completely unclear. The number of repetitions and also pain severity depends on the age of workers. Therefore, age must be in X-axis. Therefore, the explanations and also discussion must be rewritten based on new figures.
Discussion
- Page 12, the fourth paragraph, the sentence “Age is a risk factor body pain and the number of repetitions performed per minute.” is unclear.
- Page 13, the first line needs a reference.
Conclusion
- I suggest mentioning the impact of the age of workers inflicting different levels of pain in the conclusion as well.
- Based on your findings, what are possible ergonomic interventions to reduce workers’ MSD? Explain them briefly and clearly in the conclusion.
- Please add future direction, it can be regarding the limitations of the study, or for example, the next step can be studied with directions on knowing which parts of upper limb are suffering more (e.g. shoulder, arm, forearm, wrist).
Minor comments:
Following are some examples of issues relating to grammar and typos:
Abstract: the last sentence, “It reeks of the need” sounds informal.
Page 4, the first line, replace “says” with “days”.
Page 4, Section 3, first paragraph, Line 7: Replace “And” with “Moreover”
Page 5, first paragraph, Line 1: Add “,” after “On average”
Page 6, first paragraph, Line 6: Add a space between stage and reported
Section 3.3.2. first paragraph, Line 3: Replace “prove” with “proves”
Page 8, Table B1, Replace “Thep-value” with “The p-value”
Page 10, last paragraph, Line 1o: Add “,” after “Similarly”
Page 11, Section 4, second paragraph, Line 10: Replace “And” with “Moreover”
Page 12, the first line, replace “limb.s” with “limbs.”.
Page 12, second paragraph, line 8, replace “So” with “As such or Therefore”.
Page 12, sixth paragraph, line 1, Add “,” after “Contrarily”
Page 12, seventh paragraph, line 5, Replace “thescore” with “the score”
Author Response
We thank the reviewer for taking the time to provide this peer-review.
See attached document.

Round 2
Reviewer 1 Report
The first research question is excellent and you have addressed these issues in the paper. The second question is interesting but RULA is not the right tool to address that question.
The methods are described in the paper and most ergonomists are familiar with both methods.
Some of the results are presented very clearly but some are difficult to understand (Tables 3 and 5 and the statistical results tables). Perhaps it would suffice to indicate which results are statistically significant according to which statistical methods.
Some concerns include:
- Rapid Upper Extremity Assessment (RULA) is designed to evaluate awkward postures, force and repetition in order to prioritize tasks that need intervention. All tasks that you have analyzed with RULA have been identified as needing immediate investigation and intervention. Without a better description of each task it is not clear how to design an intervention. RULA is probably not the best tool to use for this investigation because all tasks need immediate intervention.
- RULA is not designed to identify which limb is experiencing more pain. The inclusion of the lower limb in the RULA tool is to provide insight into how well the worker is grounded in performing upper body tasks. RULA methodology is focused on the upper extremity.
- Pain is not a MSD. Pain is a symptom of a MSD, but could also be a symptom of something else. Pain in the upper limb does not identify where the pain be experienced. The location of the pain is a more useful indicator of what type of MSD may be present.
- It is true, as you say, that “it is difficult to give repetitions of task performance of 260 workers.” But you have identified only six tasks that you are evaluating. These six tasks provide six exposure groups. You can analyze the movements necessary to perform each of the six tasks. Each of these tasks has a duty cycle. For example, the duty cycle for a shoveling task could include the following steps: pick up the shovel, scoop up some coal, move the shovel filled with coal to another location, empty the shovel and then start over. That is one duty cycle and the number of times that duty cycle is repeated throughout the day is the number of repetitions. This is what you have to document in an ergonomics study. It is not clear what you are counting as repetitions. It is not clear where those repetitions occur – shoulder, elbow, wrist.
- The ergonomic risk factors of repetition and force act synergistically. Working posture affects the internal biomechanical forces.
- A mean BMI of 27 with a standard deviation of 6.5 indicates that more than half of the miners were overweight or obese. These does not seem possible. Please check.
The discussion is excellent and talks about the important interactions. It is very good to bring attention to the musculoskeletal impacts of coal mining in Pakistan.